# Retinoids as Alternative Antifungal Agents Against *Candida albicans*: *In Vitro* and *In Silico* Evidence

**DOI:** 10.3390/microorganisms13020237

**Published:** 2025-01-22

**Authors:** Terenzio Cosio, Alice Romeo, Enrico Salvatore Pistoia, Francesca Pica, Claudia Freni, Federico Iacovelli, Augusto Orlandi, Mattia Falconi, Elena Campione, Roberta Gaziano

**Affiliations:** 1Department of Experimental Medicine, University of Rome Tor Vergata, 00133 Rome, Italy; terenziocosio@gmail.com (T.C.); pistoiae@uniroma2.it (E.S.P.); pica@uniroma2.it (F.P.); 2Dermatologic Unit, Department of Systems Medicine, University of Rome Tor Vergata, 00133 Rome, Italy; elena.campione@uniroma2.it; 3Department of Biology, University of Rome Tor Vergata, Via della Ricerca Scientifica, 00133 Rome, Italy; alice.romeo@uniroma2.it (A.R.); claudia.freni@uniroma2.it (C.F.); federico.iacovelli@uniroma2.it (F.I.); falconi@uniroma2.it (M.F.); 4Anatomy Pathology Institute, Department of Biomedicine and Prevention, University of Rome Tor Vergata, 00133 Rome, Italy; orlandi@uniroma2.it

**Keywords:** retinoids, *in silico* analysis, ATRA, trifarotene, tazarotene, *Candida albicans*, biofilm, yeast–hyphal switching, fungal infections

## Abstract

*Candida albicans* (*C. albicans*) is the most common pathogen responsible for a wide spectrum of human infections ranging from superficial mucocutaneous mycoses to systemic life-threatening diseases. Its main virulence factors are the morphological transition between yeast and hyphal forms and the ability to produce biofilm. Novel antifungal strategies are required given the severity of systemic candidiasis, especially in immunocompromised patients, and the lack of effective anti-biofilm treatments. We previously demonstrated that all-trans retinoic acid (ATRA), an active metabolite of vitamin A, exerted an inhibitory effect on *Candida* growth, yeast–hyphal transition and biofilm formation. Here, we further investigated the possible anti-*Candida* potential of trifarotene and tazarotene, which are the other two molecules belonging to the retinoid family, compared to ATRA. The results indicate that both drugs were able to suppress *Candida* growth, germination and biofilm production, although trifarotene was proven to be more effective than tazarotene, showing effectiveness comparable to ATRA. *In silico* studies suggest that all three retinoids may exert antifungal activity through their molecular interactions with the heat shock protein (Hsp) 90 and 14α-demethylase of *C. albicans*. Moreover, interactions between retinoids and ergosterol have been observed, suggesting that those compounds have great potential against *C. albicans* infections.

## 1. Introduction

In the last decades, the incidence of invasive fungal infections (IFI) has increased considerably, representing a global public health issue. This increase correlates with a growing number of susceptible individuals, including immunocompromised patients with untreated HIV or those undergoing chemotherapy, organ transplantation, or immunosuppressive drugs, such as corticosteroids or monoclonal antibodies treatments [1]. Broad-spectrum antibiotic therapies or invasive medical procedures have also contributed to this problem [2]. Among fungal pathogens, *Candida* represents the most frequent etiologic agent of human infections [3]. In recent years, although the majority of human mycoses are caused by *C. albicans*, other non-*albicans Candida* (NAC) species, including *C. glabrata*, *C. krusei*, *C. parapsilosis*, *C. tropicalis*, *C. lusitaniae*, *C. ciferrii*, *C. blankii* and more recently *C. auris*, are considered responsible for more than 50% of all human infections. This increase in prevalence of NAC species is likely related to the wide use of antifungal drugs, such as fluconazole, either in prophylaxis or treatment of fungal infections in clinical settings [4]. *Candida albicans* is an opportunistic fungus that can normally colonize as commensal yeast different anatomical sites such as the gut, skin and vaginal tract of most individuals. Under physiological conditions, a complex and delicate equilibrium exists among *Candida*, the host immunity and local microbial communities [5]. Any alteration of the host–fungus balance may increase susceptibility to *Candida* infection and its role as a pathogen. Clinically, candidiasis ranges from localized mucocutaneous manifestations to more severe life-threatening systemic infections. Notably, the incidence of candidemia is increasing worldwide, and it is considered the fourth most common healthcare-associated invasive infection, especially in critically ill patients in intensive care units (ICU) in many tertiary care hospitals. The estimated mortality rate for candidemia is approximately 15–35% in adults and 10–15% in neonates [4]. Various virulence factors play a key role in *Candida* pathogenicity [6]. *C. albicans*’ capability to switch from yeast-to-hyphae [7] is one of the most important virulence determinants as in the filamentous forms *Candida* can breach mucosal barriers during infection, leading to invasive disease [8]. Hyphal filaments further play a significant role in *Candida* pathogenicity as they express the *extent of cell elongation* (*ECE)1* gene, which encodes the candidalysin, a cytolytic peptide toxin, able to form pore-like structures in cellular membranes, leading to membrane damage [9,10]. Moreover, *Candida*’s ability to produce biofilms on both biotic and abiotic surfaces is another important virulence factor of *C. albicans.* In fact, within biofilms, *Candida* cells are protected from host immune defense mechanisms and antifungal drugs, often leading to therapeutic failure [11,12,13].

Biofilm formation on medical devices, such as catheters, stents and prostheses, represents the major risk factor for healthcare-associated candidemia and deep tissue *Candida* infections [14]. *Candida* biofilm-associated infections have represented till now a clinical challenge. Despite many efforts, definitive solutions for *Candida* biofilms have not been found yet. Due to the lack of a unique and shared anti-biofilm strategy, many research fields have been recently investigated in order to design novel therapeutic options to prevent and/or control *Candida* biofilms. In this regard, in recent years, retinoids have attracted interest due to their *in vitro* and *in vivo* antifungal properties [15]. Retinoids are a class of chemical agents which are derivatives of vitamin A and classified into four generations on the basis of their molecular structures (Figure 1). In detail, ATRA or tretinoin (first generation), tazarotene (third generation) and trifarotene (fourth generation) are currently used in dermatology and onco-hematology clinics. Historically, the antimicrobial properties of vitamin A are well known, although the mechanisms behind its anti-infective activity have not yet been well defined. We first demonstrated that ATRA, an active vitamin A metabolite, was able to exert antifungal activity both *in vitro*, by inhibiting the germination and hyphal growth of *A. fumigatus* and *C. albicans*, and also *in vivo* in a preclinical model of pulmonary invasive aspergillosis [16,17]. The mechanisms underlying its antifungal effect are yet unclear. Based on molecular docking approaches, we proposed that ATRA might impair conidial germination in *A. fumigatus* through its interaction with the ATP-binding site of the heat shock protein (Hsp) 90, a molecular chaperone involved in many biological processes in fungi, including cell growth, cell-cycle progression, filamentation and response to environmental stress [18,19]. In addition, in *in vitro* recent studies ATRA has also proven to inhibit *C. albicans* growth, yeast-to-hyphae dimorphic switching and biofilm production [20]. Based on this evidence, herein we explored the possible anti-*Candida* potential of two other retinoid drugs, trifarotene and tazarotene, compared to ATRA. Furthermore, *in silico* studies were carried out to identify potential retinoid drug targets in *C. albicans.*

## 2. Materials and Methods

### 2.1. Candida Strain and Growth Conditions

In all *in vitro* studies the *C. albicans* reference strain ATCC 2091 was used. The strain was grown on Sabouraud dextrose agar (Difco Laboratories, Detroit, MI, USA), containing chloramphenicol, for 24 h at 30 °C. After incubation, *Candida* cells were collected by washing the slant culture with sterile saline, counted using a Bürker chamber, and adjusted to the desired concentration.

### 2.2. Antimicrobial Compounds

Stock solutions of trifarotene (catalog no. AMBH93D58E72; Merck Life Science, Darmstadt, Germany), tazarotene or ATRA (catalog no. R2625; Sigma-Aldrich, Burlington, MA, USA) and amphotericin B (AmB) (catalog no. 1397-89-3; analytical grade powder; Sigma Aldrich) were dissolved in 50% dimethyl sulfoxide (DMSO; Sigma-Aldrich) and further diluted with culture medium RPMI 1640 at a final concentration of 2.5% DMSO (*v*/*v*). Culture medium with 2.5% DMSO was used as a negative control in all experimental points.

### 2.3. Cell Growth Rate

To assess the antifungal potential of the tested retinoids on *C. albicans* growth, in comparison with that of ATRA, 2 × 10^5^
*Candida* cells were cultured into the wells of 96-well culture plates (Thermo Scientific™ Nunc™ MicroWell™ 96-Well, Nunclon Delta-Treated, Flat-Bottom Microplate; Waltham, MA, USA) containing 100 μL of RPMI 1640 medium with 10% of fetal calf serum (FCS; catalog no. 9014-81-7; Sigma-Aldrich) in the absence or presence of each compound. Briefly, we used various concentrations of tinoids ranging from 1 mM to 0.06 mM, and corresponding to trifarotene 459.6–28.72 µg/mL, tazarotene 351.5–21.96 µg/mL and ATRA 300–18.75 µg/mL. The antifungal drug Am B, at concentrations ranging from 2.2 to 0.14 µM (2–0.12 μg/mL), was used as the positive control (Appendix A) based on our previous study [20]. A volume of 100 μL of each compound was dispensed into each well. Positive controls (*Candida* cells in 200 μL of culture medium) and negative controls (200 μL of culture medium alone or 100 μL of culture medium, plus 100 μL of each retinoid or AmB without *Candida*) were included. The plates were incubated at 30 °C for 24 h. After the incubation period, the cell concentration was measured by spectrophotometry at 510 nm, using an enzyme-linked immuno-sorbent assay (ELISA) reader. The initial concentrations of *C. albicans* were considered as the concentration at 0 time. The results are the mean ± SD of three independent experiments, each performed in triplicate, and expressed as a percentage of growth inhibition vs. control.

### 2.4. Hyphal Growth Inhibition Test

The impact of trifarotene and tazarotene on *Candida* germination and hyphal growth was also evaluated. To this end, 2 × 10^5^ yeast cells were cultured into the wells of 96-well flat bottom plates in 200 μL of RPMI 1640 medium with 10% FCS, and incubated at 37 °C in the absence or presence of different concentrations of each retinoid or AmB, as described above. *Candida* cells were examined microscopically for germ tube production and hyphal growth, using a light microscope (Olympus, Carl Zeiss, Oberkochen, Baden-Württemberg, Germany) with 40× magnification objective lenses. The images of *Candida* cultures were recorded after 3 and 24 h of incubation by the digital microscope camera.

### 2.5. Quantification of Candida Biofilm by Crystal Violet and XTT Assays

To evaluate the effects of trifarotene and tazarotene on *C. albicans* biofilm production, 2 × 10^5^
*Candida* cells were cultured in 96-well flat bottom plates in 200 μL of RPMI 1640 medium, with 10% FCS and incubated at 37 °C for 24 h in the absence or presence of various concentrations of each studied compound or AmB. Crystal violet (CV) staining and 2,3-Bis-(2-Methoxy-4-Nitro-5-Sulfophenyl)-2H-Tetrazolium-5-Carboxanilide (XTT) reduction assay were used to quantify the biomass and metabolic activity of *Candida* biofilm, respectively, as previously described [20]. We performed three independent experiments, each carried out in triplicate. The results were expressed as the arithmetic means of absorbance values ± SD. In all experiments, the absorbance values of the negative controls (wells containing no cells) were subtracted from the values of the test wells to account for any background absorbance. The total biomass was also evaluated microscopically. After CV staining, the biofilm morphology was analyzed using a light microscope (Olympus, Carl Zeiss) with 40× magnification objective lenses. The images were recorded using the accompanied digital camera.

### 2.6. Quantification of Vitality of Candida albicans Cells

The impact of trifarotene and tazarotene on *C. albicans* viability was also evaluated. To this purpose, planktonic *Candida* cells were double-stained with Calcofluor White (CW), which binds to the chitin of the fungal cell wall, regardless of the metabolic state of the fungus, and propidium iodide (PI) (catalog no. P4170; Sigma Aldrich), which penetrates the cell membranes of dying or dead cells [20]. The results, presented in histogram, are the percentage of PI-stained dead cells. At least 10 fields per slide were counted.

### 2.7. Molecular Docking Simulations

The *C. albicans* Hsp90 protein was modeled using the SWISS-MODEL web server [21]. The *Saccharomyces cerevisiae* Hsp90 protein (PDBID: 2CG9) [22] was selected as a template for modeling, based on a template search carried out by the server. The structure of the *C. albicans* 14α-demethylase was obtained from the Protein Data Bank (PDB) (PDB ID: 5FSA) [23].

Molecular docking simulations were carried out through AutoDock Vina 1.2.5 software [24] to evaluate the interaction between ATRA, trifarotene or tazarotene with the *C. albicans* Hsp90 and 14α-demethylase proteins. The interaction of Hsp90 with its co-crystallized ATP molecule (PDB ID: 2CG9) [22] was evaluated as a reference. For 14α-demethylase, re-docking was performed using its natural substrate lanosterol. The three-dimensional structures of the ATP, lanosterol and retinoids were extracted from the PubChem database in SDF format and then converted into PDB format using Open Babel software [25]. The PDB structures of ligands and receptors were then further converted to PDBQT format via MGLTools software [26]. A simulation box was defined for each protein centered on the selected binding sites. For Hsp90, a box with size x = 26.25, y = 27.38, z = 24.75 Å was defined, and 16 amino acids were selected as flexible (Glu36, Leu37, Asn40, Asp43, Asp82, Ile85, Met87, Asp91, Asn95, Leu96, Ser102, Lys105, Phe123, Val125, Phe127 and Thr174). For 14α-demethylase, a box of size x = 31.88, y = 32.25, z = 30.38 Å was defined, selecting 13 flexible residues (Phe58, Tyr64, Gln66, Tyr118, Leu121, Tyr132, Phe228, Phe233, Leu376, Ser378, Phe380, Tyr505, Ser506, Ser507 and Met508). Molecular docking simulations were performed on the ENEA CRESCO6 HPC cluster [27].

### 2.8. Molecular Dynamics Simulations

Classical molecular dynamics (MD) simulations were performed selecting as input the best complexes obtained from the previous molecular docking simulations. The ff19SB force field [28] was used for proteins, and the GAFF force field [29] for ligands. The ligands’ parameters were obtained using the ACPYPE tool [30]. The topology and coordinate files of each system, used as input for the simulations, were obtained using the tleap software implemented in the AmberTools23 program [31]. Each complex was inserted into a cubic box composed of TIP3P water molecules, setting a distance of 14.0 Å between the system and box sides and neutralizing with Na^+^ and Cl^−^ counterions. Two minimization procedures were carried out to stabilize the complexes before the simulation, each consisting of 500 steps of the steepest descent and 1500 of the conjugate gradient algorithms, first applying a constraint of 2.5 kcal/mol to each system atom and then removing the constraints. The minimized structures were thermalized using the canonical NVT ensemble and a timestep of 2.0 fs, starting from a temperature of 0 K and gradually increasing it over a simulation time of 500 ps until reaching a final temperature of 300 K. A constraint of 0.2 kcal/mol was applied to each solute atom. Subsequently, complexes were simulated for an additional 1.5 ns in an NVT ensemble, decreasing constraints as 0.02, 0.01 and 0.001 kcal/mol every 500 ps. At this point, the pressure was set to 1.0 atm, and systems were equilibrated for 500 ps at constant pressure and temperature using a timestep of 2.0 fs. Temperature and pressure were kept constant at 300 K and 1.0 atm in all simulations using Langevin dynamics [32] and the Berendsen barostat [33]. Electrostatic interactions were calculated via the Particle-Mesh Ewald (PME) method [34], while a cut-off radius of 8.0 Å was set for non-bonded interactions. Finally, 200 ns of production dynamics were performed for each complex using the AMBER22 *pmemd.cuda* module [31]. The Membrane Builder tool of the CHARMM-GUI web server [35] was used to generate topology and coordinate files for a 153 × 153 × 40 Å membrane, parametrized using the CHARMM36m force field for lipids [36]. Ligands were parametrized using the CHARMM-GUI Ligand Reader & Modeler tool [37] and the CHARMM general force field [38]. The membrane composition included 85% 3-palmitoyl-2-oleoyl-d-glycero-1-phosphatidylcholine (POPC), the major component of the *C. albicans* membrane, and 15% ergosterol [39]. Four molecules of each retinoid were randomly placed in the solvent using the Packmol program [40]. The compounds–membrane systems were inserted in a rectangular box filled with TIP3P water molecules and neutralized with 0.15 M of Na^+^ and Cl^−^ ions. To remove unfavorable interactions, systems were minimized in five runs of 2000 steps each. Initial constraints of 2.5 kcal/mol were applied on each atom, sequentially halved and ultimately removed in the final run. Minimized systems were thermalized for 250 ps at 313 K in a canonical ensemble (NVT), with a timestep of 1.0 fs and constraints of 0.5 kcal/mol on membrane and ligand atoms. The systems were further equilibrated in an anisotropic NPT ensemble at a constant pressure of 1.0 atm using the Nosé–Hoover Langevin piston method [41,42], simulating at timestep 1 for 500 ps and at timestep 2 for 2 ns. Then, a production simulation of 200 ns was run for each system. A cut-off of 12.0 Å was applied for non-bonded interactions, while the PME method [34] was used to calculate electrostatic interactions. Simulations were run using the NAMD 3 program [43].

### 2.9. Trajectory Analyses

The GROMACS 2023 program [44] was used to perform RMSD, RMSF and PCA analyses. The buried surface area (BSA) between the ligands and the 14α-demethylase heme was calculated using the AmberTools23 cpptraj tool [31]. MM/PBSA analyses [45] for the protein–ligand systems were performed using the MMPBSA.py.MPI program of AMBER. To correctly handle the ATP charge, MM/PBSA calculations for the Hsp90 were performed using the non-linear approximation algorithm and setting the dielectric constant to 4. For 14α-demethylase, MM/PBSA calculations were performed with the default linear approximation and dielectric constant of 1. Average ΔG_binding_ values and standard deviations for each protein–ligand complex were calculated by the program over a series of frames extracted at fixed time intervals from each trajectory. MM/GBSA analyses for the ligand–membrane systems were carried out using the NAMD 3 [43] and MolAICal programs [46].

Ligands insertion within the membrane was monitored using a Tcl script in VMD [47] and in-house scripts in Python. Membrane thickness and lipid interdigitation were evaluated using the VMD MEMBPLUGIN tool [48]. Images were realized using the VMD program [47].

### 2.10. Statistical Analysis

Statistical analysis of data sets derived from the growth, biofilm biomass, metabolic activity and PI staining of *Candida* was carried out using GraphPad Prism 10.2.0 based on one-way analysis of variance (ANOVA) to compare each treatment with the control. Two-way ANOVA or chi-square tests were used to determine statistically significant differences among the tested molecules. Data were reported as the means ± SD of three independent experiments carried out in triplicate. The significance level for P values was considered as * *p* < 0.05; ** *p* < 0.01; *** *p* < 0.001; **** *p* < 0.0001.

## 3. Results

### 3.1. Effects of Retinoids on C. albicans Growth

In our previous work, ATRA was found to exert an inhibitory activity against the growth and biofilm formation of *C. albicans* in a dose-dependent manner [20]. In the present study, we investigated the effectiveness of trifarotene and tazarotene, compared to ATRA, in inhibiting *Candida* growth. As shown in Figure 2, both the drugs at concentrations ranging from 1.0 to 0.5 mM, significantly (one-way ANOVA, *p* < 0.0001) inhibited *Candida* growth, with 1 mM resulting as the most effective concentration, inducing a reduction by 88% in fungal growth, similar to that achieved with ATRA (90%). At a concentration of 0.5 mM, trifarotene significantly inhibited *Candida* growth in comparison with tazarotene (82% and 70%, respectively) (two-way ANOVA, *p* < 0.001) and, interestingly, its efficacy was comparable to that of ATRA (85%), although at 0.25 mM the efficacy of ATRA was significantly higher than that of trifarotene (50% and 35% inhibition, respectively) (two-way ANOVA, *p* < 0.0001). Overall, the results suggest that at concentrations of 1.0 and 0.5 mM, the antifungal activity of trifarotene in inhibiting *Candida* growth was higher than that of tazarotene and fully comparable to that of ATRA.

### 3.2. Impact of Retinoids on Yeast-to-Hyphae Dimorphic Switching in C. albicans

We have evaluated also the impact of trifarotene and tazarotene on *C. albicans* dimorphic transition between yeast and hyphal growth forms since this process represents a critical step in biofilm formation [49,50]. The microscopic images of *Candida* cultures reported in Figure 3A show that, at the earliest time point (3 h), while *Candida* control cells developed appreciable germ tubes, the treatment with trifarotene and tazarotene at the highest tested concentration (1.0 mM) was able to completely block *C. albicans* germination, similar to ATRA. However, while 0.5 mM trifarotene and ATRA could still inhibit *Candida* germination, the same concentration of tazarotene failed to block germ tube formation. Low concentrations of trifarotene (0.25 and 0.12 mM) appeared not to be able to block yeast–hyphal dimorphic transition, as documented by the visible development of germ tubes, whereas this phenomenon was found to be strongly inhibited by exposure to 0.25 or 0.12 mM ATRA. Of note, as reported in Figure 3B, the inhibitory effect of 1.0 and 0.5 mM trifarotene or 1.0 mM tazarotene on *Candida* germination was maintained even at a late time (24 h after treatment), with respect to untreated control cells that grew as long filamentous hyphae.

### 3.3. Effects of Retinoids on C. albicans Biofilm Biomass

We also evaluated the potential anti-biofilm effects of trifarotene and tazarotene on *C. albicans* biofilm, compared to ATRA, in terms of total biomass by CV staining assay. Data shown in Figure 4 indicate that at concentrations ranging from 1.0 mM to 0.25 mM both retinoids significantly inhibited the biofilm biomass production by *C. albicans* compared with controls (one-way ANOVA; *p* < 0.0001). In detail, at the highest concentration of 1.0 mM, no statistically significant differences were found between the anti-biofilm activity of trifarotene, tazarotene and ATRA. However, concentrations lower than 1.0 mM (between 0.5 and 0.25 mM) of trifarotene proved to be significantly more effective than the same concentrations of tazarotene (two-way ANOVA; *p* < 0.001), but less effective than the same concentration of ATRA at 0.25 mM (two-way ANOVA; *p* < 0.001). Moreover, ATRA, even at a low concentration of 0.12 mM, significantly inhibited *Candida* biofilm biomass compared with the same concentrations of trifarotene and tazarotene (two-way ANOVA; *p* < 0.001), which appeared to have no effect on biomass production.

### 3.4. Effects of Retinoids on C. albicans Biofilm Metabolic Activity

Based on the aforementioned results, the impact of trifarotene and tazarotene on the metabolic activity of *Candida* biofilm was also studied by XTT reduction assay. Data reported in Figure 5 show that at concentrations ranging from 1.0 mM to 0.25 mM, both retinoids significantly reduced the metabolic activity of *Candida* biofilm, while at 0.12 mM only trifarotene, similarly to ATRA, maintained a significant inhibitory activity (one-way ANOVA; *p* < 0.001) in comparison with untreated controls. The comparative analyses also indicate that at the high concentrations of 1.0 mM and 0.5 mM all the three retinoids exhibited similar antifungal efficacy. In fact, at these concentrations, no statistically significant differences were observed between trifarotene, tazarotene and ATRA (two-way ANOVA, *p* > 0.05) at 1 mM. However, 0.25 mM trifarotene and 0.25 mM ATRA showed a higher inhibitory effect than tazarotene (two-way ANOVA; *p* < 0.001 and *p* < 0.0001, respectively), whereas 0.12 mM ATRA was more effective than 0.12 mM trifarotene (two-way ANOVA; *p* < 0.001).

### 3.5. Microscopy Analysis of C. albicans Biofilm Biomass After Exposure to Retinoids

The impact of the tested retinoids on the total biomass was also evaluated by means of a light microscope after CV staining. The images reported in Figure 6 confirm that the greatest inhibitory effect on the biofilm biomass was obtained with the highest concentrations of trifarotene and tazarotene (1.0 and 0.5 mM). While at 1.0 mM no differences were found between trifarotene and tazarotene, at 0.5 mM trifarotene exhibited greater inhibition than tazarotene. Additionally, at 0.25 mM all the three retinoids showed a relatively good inhibitory effect against *Candida* biofilm biomass compared with the untreated control, although the anti-biofilm activity exerted by trifarotene and ATRA appeared to be higher than that of tazarotene. Altogether these results strongly confirm the negative impact of the three retinoids at the highest tested concentrations (1.0 and 0.5 mM) on *Candida* hyphal growth.

### 3.6. Retinoids Exhibit Dose-Dependent Fungicidal and/or Fungistatic Effect Against C. albicans

To evaluate whether the antifungal effect of trifarotene and tazarotene on *Candida* growth and biofilm formation was due to a fungistatic or fungicidal activity, we assessed the vitality of *C. albicans* cells by immunofluorescence double staining with CW and PI. Calcofluor White stains the fungal cell wall blue, regardless of its metabolic state, while PI binds to the DNA of dead cells and emits red fluorescence. The microscopic images in Figure 7 show that exposure to 1.0 mM trifarotene or tazarotene resulted in 55% and 40% of PI positive non-viable cells, respectively, vs. 60% of ATRA. By contrast, concentrations of each agent under 0.5 mM resulted in less than 2% of PI positive cells, a value very close to *Candida* control cells. Overall, the results suggest that these molecules are able to exert an inhibitory activity on the growth and biofilm formation in *C. albicans* by working either as fungicidal or fungistatic drugs, depending on the concentration used.

### 3.7. In Silico Characterization of Interactions Between Retinoids and Putative Target Proteins

Molecular docking simulations were performed to evaluate the interaction of the fungal proteins Hsp90 and 14α-demethylase with the retinoids ATRA, trifarotene and tazarotene, and to generate input structures for MD simulations. Two regions important for the functionality of the two receptors were identified as putative interaction sites: the ATP-binding pocket, located at the N-terminus of the Hsp90 protein, and the heme catalytic site of 14α-demethylase. Re-docking of the ATP ligand present in the receptor crystallographic structure was performed for Hsp90, as a reference of the protein in its active state. ATP regulates the functional cycle of Hsp90, allowing this chaperone protein to induce the folding and stabilization of protein structures. Re-docking of lanosterol, the 14 α-demethylase substrate, within its active site was also performed as a reference. Molecular docking simulations were performed in parallel using three nodes of the ENEA CRESCO6 HPC cluster [27]. Starting from the best complexes obtained from molecular docking simulations of each ligand, MD simulations were carried out to characterize the interaction of the three retinoids with the binding pockets of the two proteins (Figure 8). For each complex, a 200 ns classical MD simulation was carried out. Several analyses were performed to verify the ligands’ interactions, the complexes’ stability and the presence of alterations in the proteins’ flexibility. Simulations in the presence of ATP were used as a reference for the Hsp90 protein in its active state, while 14α-demethylase was simulated in the absence of ligands as a reference. To evaluate their binding stability, the RMSDs of each ligand within the binding pocket were calculated (Figure 9). For each trajectory, proteins were superimposed on the Cα atoms, and only movements of the ligands within the binding sites were considered for the RMSD. For Hsp90 (Figure 9A), the analysis shows that trifarotene has a higher stability within the binding site, comparable to that of the native ligand ATP. On the other hand, ATRA and tazarotene show higher RMSD values, indicating that the initial positions obtained for these two ligands via molecular docking were unfavorable compared with that of trifarotene, causing a spatial reorganization of the molecules within the binding site. For 14α-demethylase, no significant differences in stability can be detected for the three retinoids or lanosterol (Figure 9B). Since the activity of 14α-demethylase is strongly dependent on the accessibility of its heme cofactor, the buried surface area (BSA) between each ligand and the heme was evaluated. The higher this value, the larger the contact surface between the heme and the ligand. The results show that trifarotene has the highest surface of interaction with the heme, slightly higher than that of the protein substrate lanosterol. In contrast, ATRA and tazarotene show a similar trend of lower values (Figure 9C). This suggests that trifarotene has the potential to efficiently shield the active site heme from interactions with its natural substrate.

To more accurately quantify the protein–ligand interaction energies, MM/PBSA analyses [45] were carried out using AMBER software [31]. Unlike molecular docking, this technique allows the evaluation of the free energy of binding of a ligand within a binding site during an MD trajectory, also considering protein flexibility and solvent influence. MM/PBSA results are summarized in Table 1. Trifarotene shows the highest affinity for the Hsp90-binding pockets, followed by tazarotene and ATRA, showing higher or similar interaction energies if compared with the ATP cofactor. On the contrary, in 14α-demethylase all three retinoids show lower interaction energies than that of the natural substrate lanosterol.

Subsequent analyses focused on evaluating ligands’ effects on the proteins’ structural dynamics. To this aim, RMSF calculations were carried out to identify the most flexible regions in the proteins’ structures (Figure 10 and Figure 11). For Hsp90, two regions in the two monomers show altered fluctuations in the four systems (Figure 10A). As shown in the figure, this interval corresponds to a loop region near the ATP-binding pocket. In the protein active state, bound to ATP, intermediate flexibility is observed for this loop, while this region is stabilized in the presence of ATRA and subject to higher flexibility in the presence of trifarotene or tazarotene. Almost the same effect is observed for both monomers and a greater effect in the presence of trifarotene. Principal component analysis (PCA) [51] was carried out for each trajectory to identify the major alterations in proteins’ motions. Projections of the first eigenvector on each protein 3D structure allow a visualization of the most flexible regions, visually highlighting how the presence of trifarotene induces higher flexibility and reorganization in the regions surrounding the ATP-binding sites in comparison with the protein native state or in complex with the other retinoids (Figure 10B). For 14α-demethylase (Figure 11), alterations in flexibility can be noted from residues 280 to 475, representing helical and loop regions, also including ligands’ binding residues. An evident increase in flexibility of residues 375–425 can be observed in the presence of trifarotene or ATRA, and in residues 280 to 325 in the presence of trifarotene, tazarotene or lanosterol. In this case, the strongest effect on protein dynamics is induced by trifarotene binding, as can be observed from the projection of the first eigenvector on the proteins’ 3D structures (Figure 11B). In comparison with the unliganded or lanosterol-bound 14α-demethylase, in the presence of trifarotene the protein is more destabilized with the presence of different motions and flexible regions. In conclusion, these results suggest that all three retinoids, and particularly trifarotene, may represent favorable and stable interactors for these proteins, capable of competing with their native ligands. Moreover, all retinoids appear to influence the structural dynamics of the proteins, particularly for flexible regions located close to the compounds’ binding pockets.

### 3.8. Interactions of Retinoids with a C. albicans Membrane Model

MD simulations were performed to atomistically evaluate the effect of four molecules of trifarotene, ATRA or tazarotene when approaching a bilayer mimicking the *C. albicans* membrane. Simulation systems including ten ligand molecules were also simulated, but these results were discarded since the formation of large ligands’ aggregates prevented their insertion in the membrane. Simulations showed that trifarotene has a higher tendency to insert within the fungal membrane, with three out of four ligands inserted within the membrane after about 200 ns of simulation (Figure 12). On the other hand, only one ATRA or tazarotene molecule inserted in the bilayer during the 200 ns (Figure 12). Notably, translocation of tazarotene towards the opposite membrane leaflet is observed. The insertion of trifarotene leads to an alteration in the dynamical behavior of the membrane, decreasing membrane thickness and increasing lipids bilayer interdigitation; that is, the average number of contacts established between the upper and lower monolayer (Figure 13). These effects are not observed for the membrane–ATRA system and are less evident for the membrane–tazarotene system, probably because one inserted molecule is not sufficient to induce evident alterations. Interaction analyses indicate that only one molecule of trifarotene and ATRA tends to closely interact with ergosterol for about 50% of the simulation time, while the inserted tazarotene contacts ergosterol for only 25% of the simulation time. The other two inserted trifarotene molecules preferentially bind to phosphatidylcholines (POPC). MM/GBSA analyses were performed for all inserted molecules. For the three inserted trifarotene, ΔG_binding_ of −13.7 ± 0.2, −15.1 ± 0.2 and −15.0 ± 0.2 kcal/mol were obtained; a similar value of −11.0 ± 0.14 kcal/mol resulted for ATRA, while a higher ΔG_binding_ of −24.6 ± 0.3 kcal/mol for tazarotene. The higher affinity of tazarotene for the membrane is mostly due to its interactions with the POPC lipids rather than ergosterol. Overall, trifarotene shows a higher affinity than ATRA for the *C. albicans* membrane and particularly ergosterol, resulting in the insertion of more molecules during the same simulation time. The insertion of at least three trifarotene molecules can induce evident effects on this membrane model system, altering the normal thickness and fluidity of the fungal membrane. In a living system, these effects could be translated to a higher scale resulting in the induction of an altered and damaged membrane function, which could also affect the functionality of surrounding transmembrane proteins.

## 4. Discussion

In recent years, the increasing prevalence of fungal infections and the emergence of novel multi-drug-resistant (MDR) fungal pathogens represent a public health concern worldwide, especially for severely immunocompromised individuals [52]. Among fungi, *C. albicans* is the most common fungal pathogen responsible for both superficial and systemic mycotic infections in humans. The phenotypic switching from yeast to filamentous forms plays a crucial role in *Candida* pathogenicity due to the ability of *Candida* hyphae to penetrate both epithelial and endothelial cells, leading to life-threatening systemic infections. In addition to their invasive capacity, hyphae also contribute to the architectural stability of *Candida* biofilms on both host tissues and medical devices [53]. In fact, biofilm formation, despite being a common process occurring in all *Candida* spp., differs significantly from species to species. Notably, unlike NAC species, *C. albicans* biofilms exhibit a more complex and heterogeneous organization, characterized by yeasts, pseudo-hyphae and true hyphae, surrounded by extracellular polymeric substances (EPS), where hyphae serve as a support for yeast–pseudo-hyphal cells [53]. Within biofilms, *Candida* cells are more resistant to antifungal drugs compared with their planktonic cells. It has been estimated that biofilms are responsible for over 80% of all microbial infections [54]. Among microbes, *C. albicans* is the most common fungal pathogen responsible for biofilm-related infections in clinical settings. Furthermore, once a biofilm is formed on implanted medical devices or host tissues, it may serve as a reservoir for highly drug-resistant fungal cells responsible for recurrent infections and possible candidemia [55].

Considering the recalcitrant nature of *Candida* biofilms to antifungal treatments [56], novel and promising molecules that can effectively combat biofilm-associated *Candida* infections are necessarily required. In this regard, in recent years, retinoids, a group of natural or synthetic compounds deriving from vitamin A, have been proven to have a great antifungal potential [15]. These compounds are widely employed as topical dermatological agents in treating acne, psoriasis and other skin diseases [57,58,59,60]. Intriguingly, Campione et al. [61] demonstrated for the first time the effectiveness of tazarotene, both *in vivo* in the treatment of distal and lateral subungual onychomycosis, and *in vitro* against the growth of *C. albicans* and dermatophytes. In more recent works, ATRA has also been proven to have a remarkable antifungal activity against *A. fumigatus in vitro* and *in vivo* in a preclinical model of IPA [17]. ATRA has also been demonstrated to have a great anti-*Candida* efficacy by inhibiting *Candida* growth and biofilm formation [20]. Based on these findings, this study aimed at further evaluating the possible antifungal effectiveness of trifarotene and tazarotene on *Candida* growth, yeast–hyphae transition and biofilm formation compared to ATRA. The results suggest that both trifarotene and tazarotene, at concentrations ranging from 1.0 mM to 0.25 mM, significantly inhibited the growth of *C. albicans*; while at 1.0 mM their efficacy was comparable to that of ATRA, at 0.5 mM trifarotene was shown to be more effective than tazarotene. In addition, at high concentrations (1.0 and 0.5 mM), the efficacy of trifarotene in inhibiting *Candida* germination and biofilm formation was superior to that of tazarotene and similar to that of ATRA. However, it should be noted that at the concentration ≤0.25 mM, only ATRA was able to maintain significant anti-*Candida* activities, showing a dose-dependent inhibitory effect on all the tested parameters, whereas trifarotene at concentrations lower than 0.5 mM (0.25–0.12 mM) maintained a significant inhibitory effect only against the metabolic activity of *Candida* biofilm. Interestingly, the results also unveiled that all three retinoids could work as fungicidal or fungistatic agents against *C. albicans* as a function of the concentration used. Altogether, our data suggest that both trifarotene and tazarotene possess potential antifungal activity, although, in our experimental models, ATRA was found to be the most effective drug against *C. albicans.* Nevertheless, it is worth noting that the anti-*Candida* effectiveness of retinoids was reached by concentrations higher than those used for conventional antimycotics such as AmB. It is worth noting that all the tested retinoids, by blocking the germination and hyphal growth in *C. albicans,* may prevent not only the production of biofilm but also interfere with hyphae-associated adhesin expression. *Candida* adhesion represents the first step in biofilm formation and various *C. albicans* hypha-associated adhesins, including the agglutinin-like sequence (Als) family of proteins (i.e., ALS3, HWP1 and EFG1) are thought to play a key role in biofilm formation, both on abiotic and biotic surfaces such as the host epithelium, endothelium and extracellular matrix proteins [62,63,64]. In addition, by blocking *Candida* hyphal growth, these compounds could protect from invasive *Candida* infections through different mechanisms. Firstly, they may prevent tissue damage and invasion promoted by candidalysin, a cytolytic peptide toxin produced by the fungus exclusively in the filamentous form [9]. Secondly, while *Candida* yeasts are highly susceptible to phagocyte-mediated killing and elicit a protective T helper (Th)- 1 immune response, the pathogenetic hyphal forms are more difficult for phagocytic cells to internalize than *Candida* yeasts and also polarize Th0 into Th2 non-protective effector cells [10].

The molecular mechanisms behind the anti-*Candida* activity of these compounds are still unclear. It is conceivable that retinoids may specifically act on different molecular targets involved in single or multiple intracellular pathways critical for fungal growth, filamentation and biofilm formation. In this work, we focused on the investigation of two target proteins, the Hsp90 chaperone and the 14α-demethylase enzyme, which are essential to the fungal life cycle. Hsp90 is a highly conserved dimeric protein that plays a key role in promoting the folding and maturation of newly synthesized proteins as well as the degradation of proteins damaged by thermal stress [65]. Its activity is dependent on ATP binding within an N-terminal pocket, which allows proper dimerization and functionalization of the protein. In *C. albicans*, the Hsp90 chaperone plays a crucial role in fungal pathogenicity, orchestrating the dimorphic transition between yeast and filamentous forms by stabilizing numerous kinases in the cell wall integrity pathway, such as Pkc1, Bck1, Mkk2 and Mkc1 [66]. Moreover, Hsp90 is considered a central regulator of biofilm dispersion and drug resistance in fungal pathogens [67,68]. In a previous investigation, using molecular docking approaches, we demonstrated that ATRA could inhibit *A. fumigatus* germination by interacting with the N-terminal ATP-binding pocket of Hsp90 [17], which acts as a central regulator of conidial germination in *Aspergillus.* Therefore, we hypothesized that retinoids might also target this protein region in *C. albicans.* In addition, Hsp90 impairment in *Candida* drives transition from yeast to hyphal growth (contrary to what happens in *Aspergillus* [69,70]), suggesting that this protein may not be the only retinoid target in *C. albicans.* In this regard, we suggested the lanosterol 14α-demethylase (ERG11/Cyp51) as another potential target for retinoids in *C. albicans*. This protein is involved in ergosterol biosynthesis [71], a fundamental component of the fungal cell membrane that plays a critical role in maintaining its structure, function and fluidity. It is known that azoles exert their antifungal activity by targeting the catalytic heme active site of 14α-demethylase, hindering its enzymatic activity and ultimately compromising fungal membrane integrity [72]. Fungi can adapt to azole stress by adjusting the transcriptional levels of various genes [73,74]. In particular, under antifungal azole stress, Hsp90 and its client proteins play important roles in establishing resistant responses to azoles [75,76]. Thus, we hypothesized that retinoids may exert their antifungal activity through a combined inhibition of these two proteins. To explore this idea, we generated molecular docking complexes of the retinoids bound to the proteins’ active sites and characterized their interactions and structural influence on the proteins through 200 ns MD simulations. Our results suggest that all three retinoids can form stable complexes within the Hsp90 and 14α-demethylase selected binding sites, even though trifarotene showed the highest interaction energies and stability within these proteins. Trifarotene shows a higher ΔG_binding_ than ATP for Hsp90, and slightly lower than lanosterol for 14α-demethylase, but with a larger interaction surface with the heme. This suggests that this retinoid can fully occupy and block the active sites of these target proteins, preventing their activation and binding to native ligands. Moreover, trifarotene significantly destabilized the structure of both proteins compared with the other two retinoids and native ligands. By increasing the flexibility of specific loop regions proximal to the ligands’ binding sites, trifarotene induced noticeable alterations in the conformational dynamics of the two proteins. This could negatively affect their overall biological function, considering the intimate connection existing between protein collective motions and their biological activity [77].

The direct interaction of the three retinoids with a model fungal cell membrane was also evaluated through MD simulations. Again, trifarotene demonstrated the highest propensity to interact with and insert into the fungal membrane compared with ATRA and tazarotene, as indicated by the higher number of molecules entering the membrane over the same timeframe. Once inserted into the membrane, all three retinoids have shown the ability to interact with ergosterol; however, tazarotene showed a lower persistence of this interaction compared with trifarotene and ATRA during the simulation time. The insertion of trifarotene also resulted in a reduction in membrane thickness and a slight increase in lipid interdigitation, which indicates the penetration of the acyl chains from one leaflet into the opposite. This leads to enhanced interactions between the two leaflets, causing membrane compaction and a reduction in the overall flexibility and fluidity of the bilayer [78]. If transposed to a larger scale, such modifications could lead to a disruption of membrane architecture and impair the function of embedded transmembrane proteins that can sense changes in the surrounding lipid environment. Indeed, alterations in bilayer structural and dynamical features, such as membrane thickness and lipid packing, are known to influence membrane permeability, elasticity, and protein–membrane interactions [79,80], and can result in a decrease in membrane integrity and functionality, adversely affecting *Candida* growth, germ tube germination and, consequently, hyphae production. In conclusion, these results underscore the potential of retinoids, particularly trifarotene, as multi-target antifungal agents against *C. albicans*, highlighting their potential molecular mechanisms at an atomistic level.

Trifarotene and tazarotene are currently used as topical treatments for dermatological diseases such as acne vulgaris and onychomycosis, respectively, while ATRA, in combination with arsenic trioxide (ATO), is currently used *per os* as the first-line treatment for patients affected by promyelocytic leukaemia [15,81]. ATRA treatment has also been indicated in various forms of cancer including breast cancer and liver cancer [82,83,84]. However, it is noteworthy that the evidence that these drugs are able to exert antifungal effects at concentrations much higher than those used in clinical settings may represent the major limitation for the topic/systemic use of these compounds as antifungal agents. In recent times, nanotechnologies for drug delivery have emerged. The conjugation of drugs with nanocarriers may offer the advantage of applying the drugs directly into the infected tissues, improving drug bioavailability and, thus, ensuring a rapid therapeutic response [15]. Thus, the use of novel formulations based on nano drug delivery systems may be very promising for delivering retinoids efficiently and limiting, at the same time, their unwanted adverse effects. However, more work is needed to address these issues properly before such compounds can be translated from bench to bedside. Moreover, another limitation of the study is the use of a single *C. albicans* reference strain susceptible to the antifungal amphotericin B. Thus, further experiments including drug-resistant clinical isolates of *C. albicans* or NAC species, such as the new emerging multi-drug-resistant *C. auris*, as well as other fungal pathogens, are necessary to confirm the antifungal properties of these compounds.

## 5. Conclusions

Our results indicate that although both tazarotene and trifarotene are able to exert a strong anti-*Candida* activity, trifarotene was more effective than tazarotene, showing an efficacy comparable to that of ATRA. The molecular mechanisms behind their antifungal activity may involve the direct inhibition of the Hsp90 chaperone and 14α-demethylase, blocking the accessibility of their active sites and ultimately hindering the growth, the yeast-to-hyphae transition and the biofilm production of *C. albicans*. Moreover, trifarotene insertion in the fungal membrane and its binding to ergosterol can influence membrane compactness and result in a reduction of cell membrane fluidity and stability. These findings suggest that retinoids, either alone or combined with current conventional antifungal drugs, could represent very promising compounds to design novel therapeutic or preventive antifungal strategies against *C. albicans*.

## 6. Patents

The content is the object of Italian Patent Application No. 102024000028626 filed on 16 December 2024.

## Figures and Tables

**Figure 1 microorganisms-13-00237-f001:**
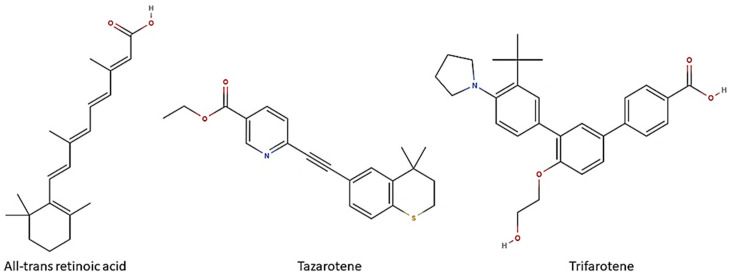
Retinoids and their chemical structure. All-trans retinoic acid (ATRA) or tretinoin are obtained by chemically modifying the polar groups of vitamin A; tazarotene is a polycyclic molecule, resulting from the cyclization of a side chain; trifarotene, a synthetized fourth-generation retinoid, belongs to the class of terphenyls with a structure containing the 1,3-diphenylbenzene skeleton. Created using molView^®^ (https://molview.org/, accessed on 25 November 2024).

**Figure 2 microorganisms-13-00237-f002:**
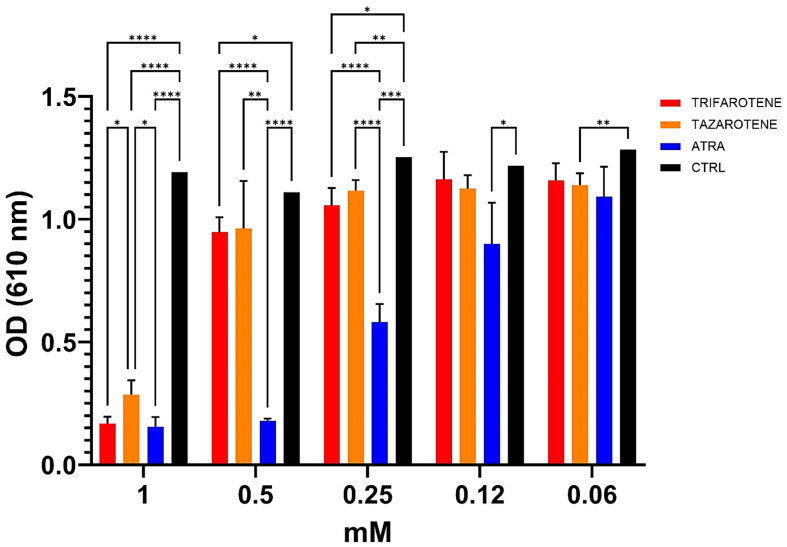
Inhibitory activity of retinoids on *C. albicans* growth. *Candida* yeasts were cultured for 24 h at 30 °C in the absence or presence of different concentrations of trifarotene, tazarotene or ATRA. The drug AmB was used as a positive control. Results are the mean ± SD of three independent experiments, each performed in triplicate, and expressed as a percentage of growth inhibition vs. control: 0.5 mM trifarotene vs. 0.5 mM tazarotene *p* < 0.001; 0.25 mM ATRA vs. 0.25 mM trifarotene and 0.25 mM tazarotene *p* < 0.001. Two-way ANOVA, * *p* < 0.05; ** *p* < 0.01; *** *p* < 0.001; **** *p* < 0.0001.

**Figure 3 microorganisms-13-00237-f003:**
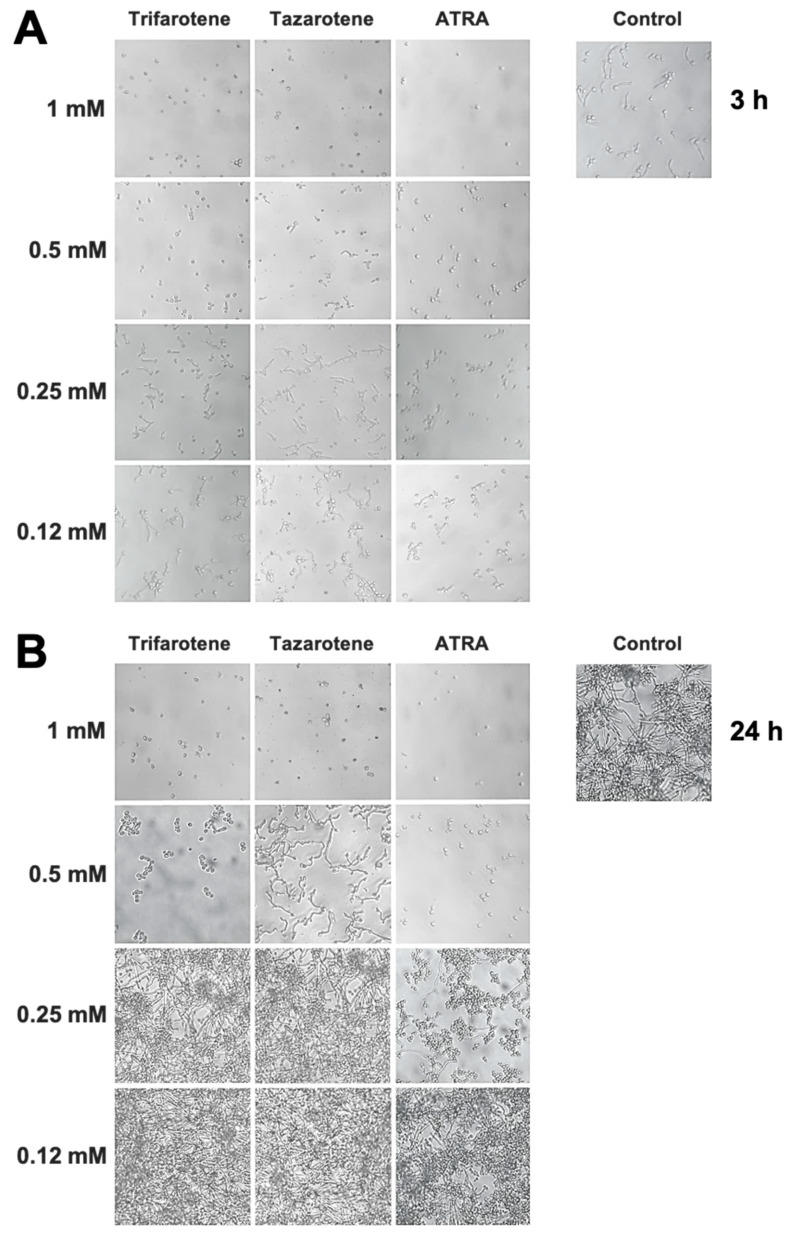
Inhibitory effect of retinoids on *C. albicans* dimorphic transition. *Candida* yeasts were incubated at 37 °C in the absence or presence of different concentrations of trifarotene, tazarotene or ATRA. AmB was used as a positive control drug. The dimorphic transition from yeast to hyphae was analysed microscopically after 3 (**A**) and 24 h (**B**) of incubation. One representative experiment of three is shown.

**Figure 4 microorganisms-13-00237-f004:**
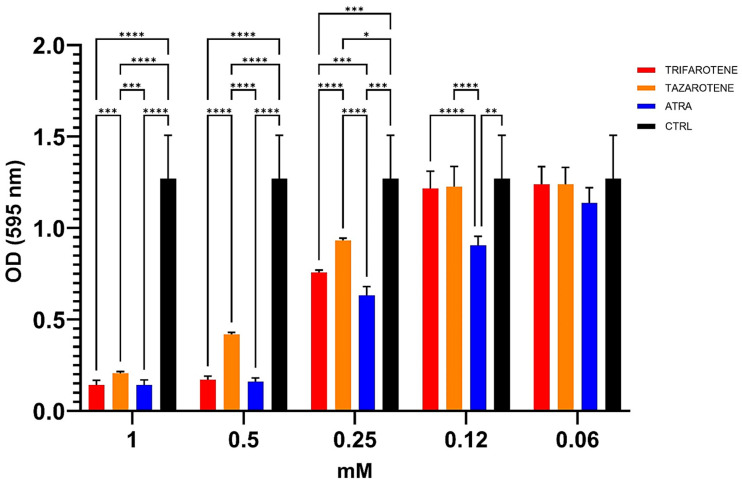
Inhibitory effects of retinoids on *C. albicans* biofilm biomass production. *Candida* yeasts were cultured for 24 h at 37 °C with or without dfferent concentrations of trifarotene, tazarotene or ATRA. The drug AmB was used as a positive control drug. Biofilm production as biomass (A) was evaluated by CV assay. The absorbance intensity of CV dye was measured by spectrophotometry at 595 nm. Results are the means ± SD of three independent experiments, each performed in triplicate: 0.5 mM trifarotene and 0.5 mM ATRA vs. 0.5 mM tazarotene *p* < 0.0001; 0.25 and 0.12 mM ATRA vs. 0.25 and 0.12 mM trifarotene *p* < 0.001; 0.25 mM trifarotene vs. 0.25 mM tazarotene *p* < 0.0001. Two-way ANOVA, * *p* < 0.05, ** *p* < 0.01; *** *p* < 0.001; **** *p* < 0.0001.

**Figure 5 microorganisms-13-00237-f005:**
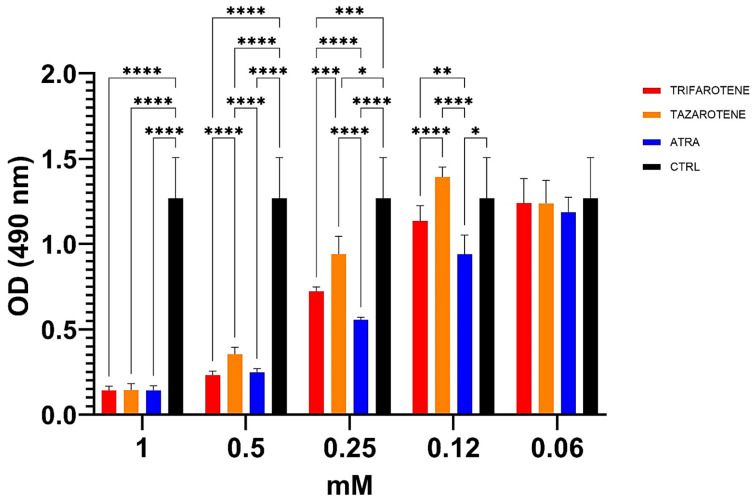
Inhibitory activity of retinoids on metabolic activity of *C. albicans* biofilm. *Candida* yeasts were cultured for 24 h at 37 °C with or without different concentrations of trifarotene, tazarotene or ATRA. The antifungal AmB was used as a positive control. XTT reduction assay was used to evaluate the metabolic activity of *Candida* biofilm, by using a spectrophotometer plate reader at 490 nm. Results are the means ± SD of three independent experiments, each performed in triplicate: 0.25 mM trifarotene and 0.25 mM tazarotene *p* < 0.01; 0.25 mM ATRA vs. 0.25 mM tazarotene. Two-way ANOVA, *p* < 0.001. * *p* < 0.05, ** *p* < 0.01; *** *p* < 0.001; **** *p* < 0.0001.

**Figure 6 microorganisms-13-00237-f006:**
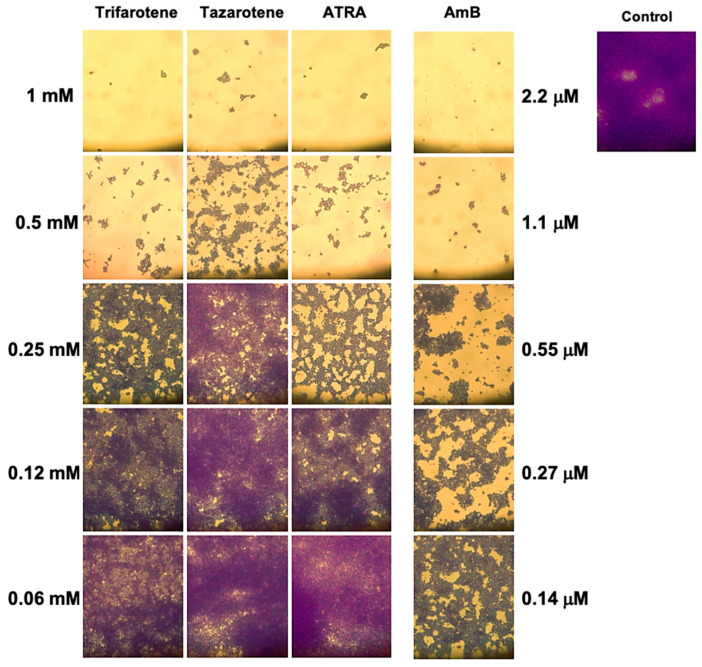
Evaluation of *C. albicans* biofilm biomass after exposure to retinoids by microscopy. After CV staining, the total biofilm biomass was analyzed by a light microscope with 40× magnification objective lenses.

**Figure 7 microorganisms-13-00237-f007:**
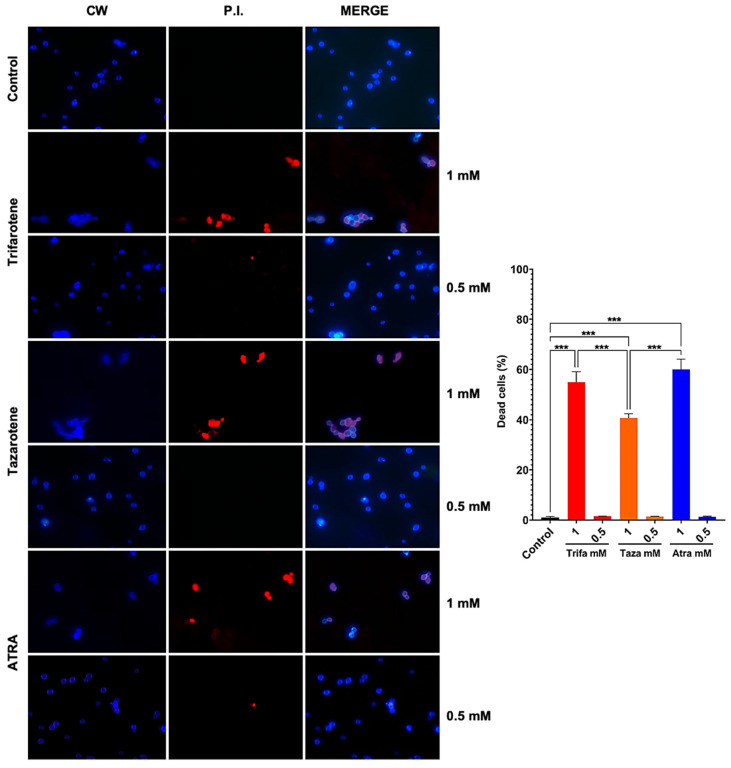
*Candida albicans* vitality was evaluated by double staining with Calcofluor White (CW) and propidium iodide (PI). *C. albicans* planktonic yeast cells were incubated at 30 °C for 24 h with or without trifarotene, tazarotene or ATRA (1–0.06 mM). After incubation, *Candida* cells were double-stained with CW and PI. Fluorescent microscopy images of *Candida* cells are shown; blue fluorescence represents the fungal cell wall stained by CW (left column); red fluorescence (middle column) represents nucleic acids of dead cells stained by PI; merged images (in the right column), show cells double labeled with CW + PI. All images were recorded using a fluorescence microscope at 100x magnification. The results, presented in histogram, are the percentage of dead cells. At least 10 fields per slide were counted. Chi-square test: *** *p* < 0.001.

**Figure 8 microorganisms-13-00237-f008:**
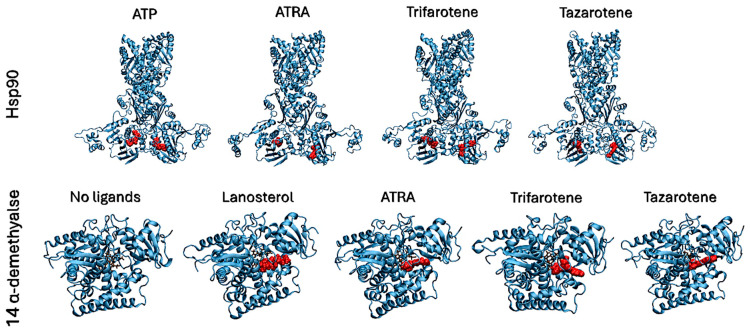
Best complexes obtained from molecular docking simulations and used as inputs for MD simulations. Proteins are shown as cyan cartoons, ligands are shown as red spheres. The 14α-demethylase heme is shown in brown as sticks.

**Figure 9 microorganisms-13-00237-f009:**
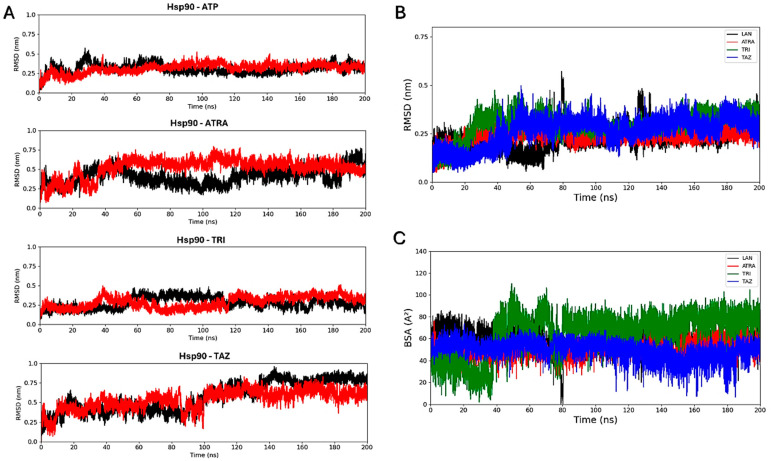
Root mean square deviation analyses. RMSD calculated for each ligand within the Hsp90 dimer, with the two monomers indicated in black and red (**A**). Ligands’ RMSDs calculated for the 14α-demethylase systems (**B**). Buried surface area (BSA) calculated between the 14α-demethylase heme and each ligand (**C**). ATP, Adenosine triphosphate; ATRA, all-trans retinoic acid; LAN, lanosterol; TAZ, tazarotene; TRI, trifarotene.

**Figure 10 microorganisms-13-00237-f010:**
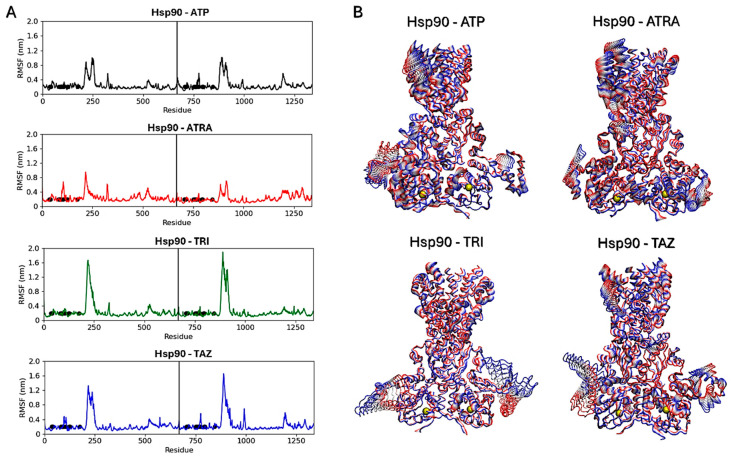
Hsp90 dimers RMSF analyses. Black points indicate residues involved in the ligands’ binding. The vertical lines indicate the separation between the two monomers (**A**). Projections of the first eigenvectors obtained from PCA analyses on the proteins’ 3D structures. Proteins are shown as tubes, and colors from red to blue indicate the proteins’ motions from the beginning to the end of the simulations. The ligands’ locations are indicated by yellow spheres (**B**). ATP, Adenosine triphosphate; ATRA, all-trans retinoic acid; LAN, lanosterol; TAZ, tazarotene; TRI, trifarotene.

**Figure 11 microorganisms-13-00237-f011:**
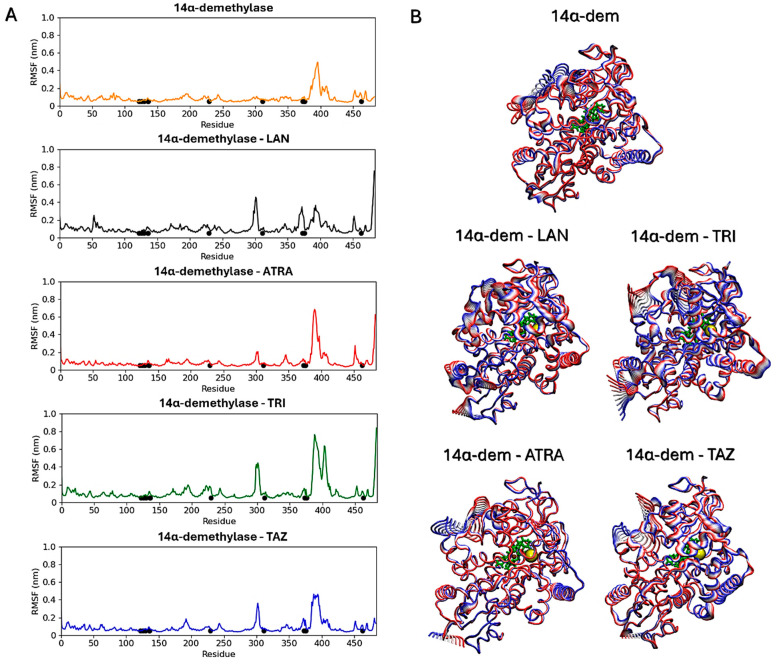
14α-demethylase RMSF analyses. Black points indicate residues involved in the ligands’ binding (**A**). Projections of the first eigenvectors obtained from PCA analyses on the proteins’ 3D structures. Colors from red to blue indicate the proteins’ motions from the beginning to the end of the simulations. The ligands’ locations are indicated by yellow spheres (**B**). ATP, Adenosine triphosphate; ATRA, all-trans retinoic acid; LAN, lanosterol; TAZ, tazarotene; TRI, trifarotene.

**Figure 12 microorganisms-13-00237-f012:**
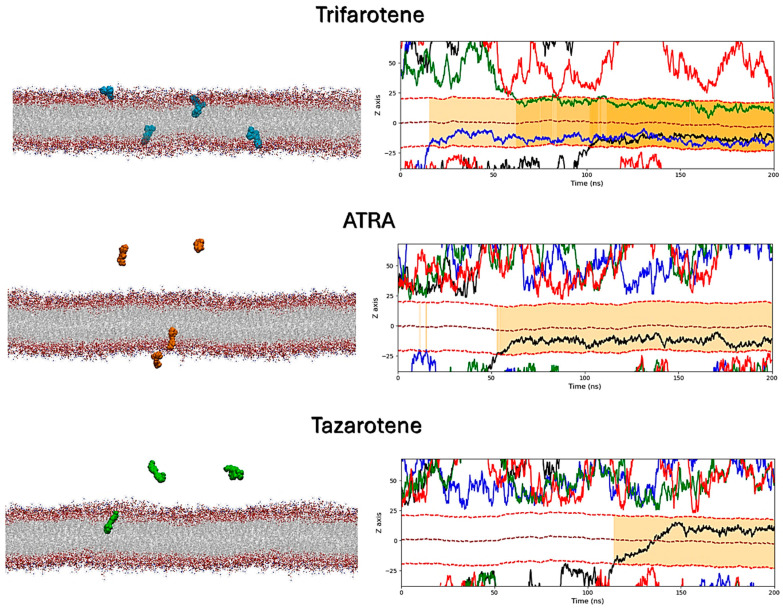
Interactions of retinoids with a *C. albicans* membrane model. On the left, the final frames of the membrane simulations show the location of trifarotene (cyan spheres), ATRA (orange spheres) or tazarotene (green spheres) molecules. The membrane is shown as sticks in gray, with the polar heads highlighted in red. On the right, the Z-axis position relative to the membrane of each trifarotene, ATRA or tazarotene molecule is represented by black, blue, green and red lines. Red dashed lines indicate the upper and lower membrane boundaries, and the brown dashed line indicates the membrane center. Ligand insertion into the membrane is represented by a yellow gradient; darker yellow indicates a greater number of inserted molecules.

**Figure 13 microorganisms-13-00237-f013:**
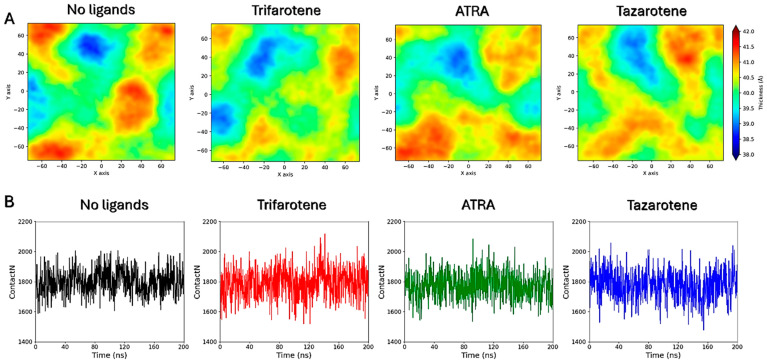
Membrane thickness maps. Membrane thickness maps calculated for the four membrane systems (**A**). Lipid interdigitation calculated as the number of contacts between the upper and lower layer for the four membrane systems (**B**).

**Table 1 microorganisms-13-00237-t001:** MM/PBSA results for the Hsp90 and 14α-demethylase proteins. ATP, Adenosine triphosphate; ATRA, all-trans retinoic acid; LAN, lanosterol; TAZ, tazarotene; TRI, trifarotene.

Protein	MM/PBSA ΔG_binding_ (kcal/mol)				
	ATP	LAN	ATRA	TRI	TAZ
**Hsp90 (monomer 1)**	−40.9 ± 5.2	-	−36.5 ± 2.4	−53.6 ± 4.2	−50.1 ± 4.6
**Hsp90 (monomer 2)**	−30.5 ± 6.4	-	−39.6 ± 2.9	−56.6 ± 4.0	−41.7 ± 4.5
**14α-demethylase**	-	−41.3 ± 3.7	−28.5 ± 4.7	−31.7 ± 5.8	−34.5 ± 4.2

## Data Availability

The original contributions presented in the study are included in the article/Appendix A, further inquiries can be directed to the corresponding author.

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
