# Peer review of "Retinoids as Alternative Antifungal Agents Against *Candida albicans*: *In Vitro* and *In Silico* Evidence"

_microorganisms, 2025, doi:10.3390/microorganisms13020237_

Round 1
Reviewer 1 Report
Comments and Suggestions for Authors
microorganisms-3405571
This Manuscript in general:
- The objectives were reached where authors found that that trifarotene and tazarotene were able to suppress Candida growth, germination and biofilm production of Candida albicans
- Methods are suitable to investigate the objectives. Moreover, they are extensive explained.
- Results and discussion are well presented
- References in all sections are adequate to the study
Here, the major comments:
1- Abstract: it is very long specially in the background. This section in the website is different from the Manuscript. The abstract in the website is more suitable than that in the MS
2- Introduction: is too long about C. albicans and their drugs specially lines from 43-89. It should be summarized in all information.
3- Statistical analysis: it is not suitable for the data, in details;
- Data in Figures 2, 4 and 5: they should be analyzed with Two-Way ANOVA because there are 2 factors (drug and concentration), thus, add letters above columns as capitals (between drugs in the same concentration) and small letters (among concentrations of the same drug). Accordingly, one figure is enough instead of 5 sub-figures.
- The previous comment for figure 7 (right part). In this one, add (A) and (B) for each one inside the figure
- Table 1: where the statistical analysis among data in the same column?
4- Plagiarism should be reduced
Author Response
Dear Reviewer,
Thank you very much for giving us the opportunity to revise our manuscript (Submission ID: microorganisms-3405571). Thanks for your valuable comments, which undoubtedly have improved the quality of our paper.
Please find below a point-to-point letter to respond to your comments and suggestions. All the changes were highlighted in red in the revised manuscript (marked copy).
Please let me know if you require any additional information on our paper.
I am looking forward to hearing from you soon.
Best regards,
Dr. Roberta Gaziano
This Manuscript in general:
- The objectives were reached where authors found that that trifarotene and tazarotene were able to suppress Candida growth, germination and biofilm production of Candida albicans
- Methods are suitable to investigate the objectives. Moreover, they are extensive explained.
- Results and discussion are well presented
- References in all sections are adequate to the study
Here, the major comments:
- Abstract: it is very long specially in the background. This section in the website is different from the Manuscript. The abstract in the website is more suitable than that in the MS
Thank you for your suggestion. We apologize for the inconvenience. In the revised version of the manuscript, the abstract has been replaced with that on the website.
- Introduction: is too long about C. albicans and their drugs specially lines from 43-89. It should be summarized in all information.
Dear Reviewer, the “Introduction” section has been shortened and some sentences, as also suggested by the other reviewer, have been moved to the “Discussion” section.
3- Statistical analysis: it is not suitable for the data, in details;
- Data in Figures 2, 4 and 5: they should be analyzed with Two-Way ANOVA because there are 2 factors (drug and concentration), thus, add letters above columns as capitals (between drugs in the same concentration) and small letters (among concentrations of the same drug). Accordingly, one figure is enough instead of 5 sub-figures.
We used the Student's t-test for the statistical analysis of the data reported in Figure 2, 4, and 5 as we compared two by two the effects of each concentration of trifarotene or tazarotene vs the same concentration of ATRA; and two by two the effects of each concentration of trifarotene vs the same concentration of tazarotene. For instance, 1 mM of trifarotene vs 1 mM of ATRA; 1 mM of tazarotene vs 1 mM of Atra; 1 mM of tazarotene vs 1 mM of trifarotene.
- The previous comment for figure 7 (right part). In this one, add (A) and (B) for each one inside the figure
Thank you for the suggestions. Concerning this point, we think that the visualization of the effects of the tested retinoids on different parameters of fungal growth can help readers better comprehend them. Consistently, in all the experiments we have used ATRA concentrations as the positive control for studying the antifungal activity of the other two retinoids.
- Table 1: where the statistical analysis among data in the same column?
Thank you for pointing this out. However, in our study, we conducted a single simulation for each protein-ligand complex due to the high number of systems simulated and their computational cost, limiting our ability to perform statistical tests to evaluate the differences in ligands free energies of binding. The mean and standard deviation values reported in Table 1, for each protein-ligand complex, were calculated by the program from the same simulation by averaging the ΔG binding of trajectory frames extracted at fixed time intervals. This has also been specified in the text (see lines: 246-248).
4- Plagiarism should be reduced
Dear Reviewer, we have modified and rephrased some sentences to reduce the plagiarism.
Finally, we have appreciated all of your feedback and have carefully considered your suggestions for improving our manuscript.
Reviewer 2 Report
Comments and Suggestions for Authors
microorganisms-3405571-peer-review-v1
This is an interesting research project, with potential applications in combat versus Candida albicans. Authors have constructed interesting experimental and modeling research project and in my opinion paper can be suggested for publication, however, some adjustments and correction needs to be made by the authors.
Ln3: Some of the MDPI journal do not require that in vitro, in silico etc., needs to be in italics. This is the internal rules. Maybe I am bit conservative, regarding this issue, and all words that not in English, I believe that needs to be in italics. However, if you have mentioned it in vitro in italics, then, in silico needs to be in italics as well. Please, check the entire manuscript for similar adjustments.
In the affiliation information, initials for all authors need to be adjusted, and add "." after the initials. Example E.P.; A.O., etc.
The introduction is very informative, but very long. I am appreciating force of authors to provide all this information, but for the purposes of the research paper, maybe will be appropriate to reduce a bit introduction and move several parts to the discussion section.
Any specific reason authors used 30C as incubation temperature (Ln147). Since in the introduction was several times mentioned importance of Candida albicans as relevant pathogen, maybe will be more appropriate that test will be conducted at 37C. Please, provide justification for use of this specific temperature? Are there some preliminary results indicating this temperature as more relevant for the experimental procedures? Furter in the manuscript, several other experiments were conducted at 37C.
Please, for all suppliers of reagents and equipment provide information including name of the company, address: city, state (in case of federal country) in abbreviated way, and country. Try to provide the headquarters address of the company and not a local distributor. Example, Ln52 Sigma Aldrich is not Italian company. After ones introduced, in following occasions, use only name of the company, without other information regarding the address.
Most of the Material and Methods were described with reasonable details.
Please microscopic conditions needs to be presented better in the material and methods section.
In my opinion, the discussion section can be extended. In addition, moving some parts from the introduction to the discussion, authors can pay attention to have all experimental parts discussed appropriately into the Discussion. Special attention need to be given to the first part of the experimental procedures. Moreover, the modelling part of the research is very interesting, and in fact this is more important part of the paper, but as well, maybe a bit more attention needs to be given to this part. Authors needs to try to have a better balance between different parts of the manuscript, and definitely increase bit discussion section, and enrich with more details and comparing to similar research projects.
Please, references need additional attention, Example some journal names are abbreviated, and other presented fully. Please, check the instructions for Microorganisms and adjust the reference list.
Maybe help from more experience colleague will be good option in formatting and adjusting the content of the manuscript.
Author Response
Dear Reviewer,
Thank you very much for allowing us to revise our manuscript (Submission ID: microorganisms-3405571). Thanks for your valuable comments that undoubtedly have improved the quality of our paper.
Please, you will find below a point-to-point letter to respond to your comments and suggestions. All the changes were highlighted in red in the revised manuscript (marked copy).
Please let me know if you require any additional information on our paper.
Looking forward to hearing from you soon.
Best regards,
Dr. Roberta Gaziano
This is an interesting research project, with potential applications in combat versus Candida albicans. Authors have constructed interesting experimental and modeling research project and in my opinion paper can be suggested for publication, however, some adjustments and correction needs to be made by the authors.
Thank you very much for your interest and valuable comments and suggestions to improve our manuscript.
Ln3: Some of the MDPI journal do not require that in vitro, in silico etc., needs to be in italics. This is the internal rules. Maybe I am bit conservative, regarding this issue, and all words that not in English, I believe that needs to be in italics. However, if you have mentioned it in vitro in italics, then, in silico needs to be in italics as well. Please, check the entire manuscript for similar adjustments.
Thank you for pointing this out. All words that are not in English have been modified accordingly as per your suggestion.
In the affiliation information, initials for all authors need to be adjusted, and add "." after the initials. Example E.P.; A.O., etc.
Thank you for your suggestion. Affiliations have been modified accordingly.
The introduction is very informative, but very long. I am appreciating force of authors to provide all this information, but for the purposes of the research paper, maybe will be appropriate to reduce a bit introduction and move several parts to the discussion section.
We agree with this comment. The “Introduction” section has been shortened and some sentences have been moved to the “Discussion” section.
Any specific reason authors used 30C as incubation temperature (Ln147). Since in the introduction was several times mentioned importance of Candida albicans as relevant pathogen, maybe will be more appropriate that test will be conducted at 37C. Please, provide justification for use of this specific temperature? Are there some preliminary results indicating this temperature as more relevant for the experimental procedures? Furter in the manuscript, several other experiments were conducted at 37C.
We used the incubation temperature of 30°C because the optimal growth temperature for C. albicans ranges between 28-30°C. At 30°C, the fungus grows as yeast cells. Incubation temperature of 37°C, in addition to the presence of 10% of FCS in the culture medium, mimics a stress condition for the fungus, promoting in C. albicans the germ tube formation and filamentation (Please see the article DOI: 10.4236/ojmm.2013.33028.) For this reason, to study the impact of retinoids on Candida hyphal growth and biofilm formation, we specifically used the incubation temperature of 37°C.
Please, for all suppliers of reagents and equipment provide information including name of the company, address: city, state (in case of federal country) in abbreviated way, and country. Try to provide the headquarters address of the company and not a local distributor. Example, Ln52 Sigma Aldrich is not Italian company. After ones introduced, in following occasions, use only name of the company, without other information regarding the address.
For all reagents and equipment, we have provided details regarding the company and the related address as per your suggestion.
Most of the Material and Methods were described with reasonable details. Please microscopic conditions needs to be presented better in the material and methods section.
A more detailed description regarding the microscopic analysis performed to visualize C. albicans hyphal growth (Lines,,,), as well as the morphology of biofilm ( lines…), has been provided in the “Material and Methods” section.
In my opinion, the discussion section can be extended. In addition, moving some parts from the introduction to the discussion, authors can pay attention to have all experimental parts discussed appropriately into the Discussion. Special attention need to be given to the first part of the experimental procedures. Moreover, the modelling part of the research is very interesting, and in fact this is more important part of the paper, but as well, maybe a bit more attention needs to be given to this part. Authors needs to try to have a better balance between different parts of the manuscript, and definitely increase bit discussion section, and enrich with more details and comparing to similar research projects.
Dear Reviewer thank you for your suggestion. The “Discussion” section has been expanded. Some sentences have been moved from the introduction to the Discussion as per your suggestion. Moreover, a broader discussion about the results obtained from in silico studies has been provided.
Please, references need additional attention, Example some journal names are abbreviated, and other presented fully. Please, check the instructions for Microorganisms and adjust the reference list. Maybe help from more experience colleague will be good option in formatting and adjusting the content of the manuscript.
All the references have been formatted according to the Journal guidelines.
Finally, we have appreciated all of your feedback and have carefully considered your suggestions for improving our manuscript.
Round 2
Reviewer 1 Report
Comments and Suggestions for Authors
Dear Authors
thanks for reaching my comments, but still these 2 comments
1- Data in Figures 2, 4 and 5: they should be analyzed with Two-Way ANOVA because there are 2 factors (drug and concentration), thus, add letters above columns as capitals (between drugs in the same concentration) and small letters (among concentrations of the same drug). Accordingly, one figure is enough instead of 5 sub-figures.
2- Plagiarism is 28%, it should be reduced
Author Response
Thank you very much for allowing us to revise our modified manuscript (Submission ID: microorganisms-3405571).
We also would like to thank the reviewers for his/her valuable comments. You will find a point-to-point letter to respond to the reviewer’s comments and suggestions. All the changes made are in red font (marked copy).
1-Data in Figures 2, 4 and 5: they should be analyzed with Two-Way ANOVA because there are 2 factors (drug and concentration), thus, add letters above columns as capitals (between drugs in the same concentration) and small letters (among concentrations of the same drug). Accordingly, one figure is enough instead of 5 sub-figures.
1-Dear Reviewer, we have re-analysed all the obtained data and based on your suggestions, we have modified the Figures. We hope that the presentation of the figures has been improved and clearer to the reader.
2- Plagiarism is 28%; it should be reduced.
2-Thank you for pointing out this issue. We have rechecked the paper to reduce all the plagiarism in the main text, based on iThenticate report on the website.
Finally, we appreciate all of your feedback and have carefully considered your suggestions for improving our manuscript.
